# Macronutrient Intake in Soccer Players—A Meta-Analysis

**DOI:** 10.3390/nu11061305

**Published:** 2019-06-09

**Authors:** Michal Steffl, Ivana Kinkorova, Jakub Kokstejn, Miroslav Petr

**Affiliations:** Faculty of Physical Education and Sport, Charles University, 162 52 Prague 6, Czech Republic; kinkorova@ftvs.cuni.cz (I.K.); kokstejn@ftvs.cuni.cz (J.K.); petr@ftvs.cuni.cz (M.P.)

**Keywords:** nutrition, protein, carbohydrates, lipids, association football

## Abstract

The nutrition of soccer players is an important topic and its knowledge may help to increase the quality of this popular game and prevent possible health problems and injuries in players. This meta-analysis aims to estimate the current dietary trends of three basic macronutrients in junior and senior soccer players during the first two decades of the 21st century. We analyzed data from 647 junior players (mean age 10.0–19.3) from 27 groups, and 277 senior (mean age 20.7–27.1) players from 8 groups from altogether 21 papers in this meta-analysis. Weighted averages were calculated for each macronutrients. Protein intake is higher than recommended in both juniors, 1.9 95% confidence interval (CI) 1.8–2.0 g/kg/day, and seniors 1.8 95% CI 1.6–2.0 g/kg/day. However, carbohydrate intake is still below the recommended values in both groups (5.7 95% CI 5.5–5.9 g/kg/day in junior and 4.7 95% CI 4.3–5.0 g/kg/day in senior players). The proportion of fat as total energy intake is in concordance with the recommendations (31.5 95% CI 32.0–35.9% in junior and 33.1 95% CI 29.9–36.2% in senior players). In particular, due to possible health complications, the small carbohydrate intake should be alarming for coaches, nutritional experts, and parents.

## 1. Introduction

Association football, commonly known as football or soccer is the most popular sport worldwide [1]. Most likely, it is also one of the most dynamically developed sports, which puts high demands on individual physical performance [2]. It is well-known that nutrition plays an important role in the training process. Properly led training should include recommendations regarding macronutrients intake based on knowledge verified by research. Athletes often in an effort to improve performance use a variety of nutritional ergogenic aids to enhance performance. Often, however, it would be sufficient only to adjust the standard diet to meet their requirements. Even though several recommendations have been made over the past twenty years, current reviews indicated that macronutrient intake in soccer is probably still not adequate to fulfil the requirements of players [3,4]. To better estimate the macronutrient intake, we decided on creating a meta-analysis based on the scientific works published during the first two decades of the 21st century. Our results may serve as an example for people who are responsible for successful training programs in soccer.

Although macronutrients could not be counted directly among the ergogenic aids, which are classified as nutritional, pharmacologic, physiologic, or psychologic and range from the use of accepted techniques such as carbohydrate loading to illegal and unsafe approaches such as anabolic-androgenic steroid use [5], macronutrients could have an ergogenic effect. Most nutritional aids can be categorized as a potential energy source, an anabolic enhancer, a cellular component, or a recovery aid. Studies have shown that carbohydrates consumed immediately before or after exercise enhance performance by increasing glycogen stores and delaying fatigue. Proteins may serve an anabolic role by optimizing body composition [6,7]. Moreover, nutrition plays an important role in the normal growth and development, maintaining health and well-being, reducing the risk of illness and injury; optimal diet may help to avoid the possible health complications [8]. Aside from decreased performance, the restrained eating can cause significant detrimental outcomes to body function [9].

The topic of macronutrients in soccer has been studied for many years. Since the eighties, discussion has centered mainly on the higher carbohydrate (CHO) daily intake [10,11,12,13,14,15]. In addition to CHO, the focus has also been on protein and lipid intake [16,17,18]. A number of nutrition recommendations for soccer players have ever since evolved. In 1994, 7–10 g of CHO per kg body mass was suggested for maximizing the glycogen storage [19]. In 2011, 5–10 g/kg/day of CHO was recommended [20]. The American Dietetic Association (ADA) recommended 6–10 g/kg/day of CHO in 2009 [21] as well as in 2016 [22]. According to the International Society of Sports Nutrition (ISSN), during high volume exercise the endogenous glycogen stores are maximized by following a high carbohydrate diet (8–12 g/kg/day) [23]. In 1994, it was stated that a protein intake of 1.4–1.7 g/kg/day should be adequate for soccer players [16]. The ADA recommended a protein intake of 1.2–1.7 g/kg/day in 2009 [21], and according to the ADA in 2016, the dietary protein intake necessary to support metabolic adaptation and for protein turnover (repair, remodeling) should range from 1.2 to 2.0 g/kg/day [22]. ISSN recommend for the majority of exercising individuals to consume approximately 1.4 to 2.0 g/kg/day of protein to optimize exercise training induced adaptations [24]. According to the ADA, fat intake should range from 20% to 35% of total energy intake (TEI) [21] and similarly the ISSN recommends 25–35% [7].

Selecting effective dietary strategies for elite football players requires comprehensive information on their dietary intake. Because of the relatively small amount of information on macronutrient intake in soccer from individual studies, there is a need to carry out a meta-analysis of developments. We therefore decided to carry out a meta-analysis describing the current situation in soccer players’ macronutrient intake and developments in this field over the past twenty years. This meta-analysis aims (1) to calculate the macronutrients intake in soccer players from elite and sub-elite leagues based on the results of individual observation studies; (2) to estimate the trends of macronutrients intake in the 21st century in soccer players; (3) to compare macronutrients intake between junior and senior soccer players; and finally (4) to compare the intake of the basal macronutrients—proteins, CHO, and fat with recommendation values.

## 2. Methods

We conducted this meta-analysis in accordance with the Preferred Reporting Items for Systematic Reviews and Meta-Analyses (PRISMA) statement [25].

### 2.1. Search Strategy

We identified relevant papers published between the years 2000 and 2019 referenced in EBSCOhost—SPORTDiscus, Scopus, and PubMed. We used the same search terms in all the databases (Table 1). Additionally, we hand-searched the reference lists of eligible papers and of several recently published reviews for further studies.

### 2.2. Inclusion Criteria

We assessed the dietary habits of male soccer players in two different age categories: juniors and seniors. We only included cross-sectional studies or baseline data from intervention trials that met the following inclusion criteria: date of publication 2000–2019, written in English language, with the presence of information about macronutrient intake (proteins, CHO, lipids, or fat). We excluded studies focused on the effect of special supplementations of macronutrients and micronutrients, where no information was given about dietary habits before the intervention.

### 2.3. Data Collection and Quality Assessment

We downloaded all the potential papers in EndNote, and deleted all the duplicates. We then screened titles and abstracts of papers to identify studies that potentially met the eligibility criteria. Full texts were subsequently assessed for eligibility. After examining the full text for inclusion or exclusion, the remaining papers were included in the synthesis. Additionally, we added two relevant papers that were identified from reference lists of the papers identified through database searching. After identification of eligible papers, the Joanna Briggs Institute–Qualitative Assessment and Review Instrument (JBI-QARI) was used to assess the methodological quality of the included studies [26]. All the authors were in agreement about the quality of all papers.

### 2.4. Calculation of Weighted Averages Nutritional Intake

As main measures, we calculated the weighted averages for junior and senior players separately. We used a fixed model approach where the weighted average was calculated as $\bar{T.}=\frac{\sum_{i=1}^{k}w_{i}T_{i}}{\sum_{i=1}^{k}w_{i}}$, where w_i_ is the weight of each study calculated as $w_{i}=\frac{1}{v_{i}}$, and v_i_ is the within-study variance for study (i) and T_i_ is mean for study (i). The variance of the weighted average is defined as $v.=\frac{1}{\sum_{i=1}^{k}w_{i}}$ and standard error $SE\left(\bar{T.}\right)=\sqrt{v.}$. The 95% confidence interval (CI) for weighted average was computed as lower limit = $\bar{T.}-1.96*SE(\bar{T.}$) and upper limit = $\bar{T.}+1.96*SE(\bar{T.}$) [27]. Additionally, we estimated the heterogeneity using Cochran Q statistic as $Q=\overset{k}{\underset{i=1}{\sum }}w_{i}{\left(T_{i}-\bar{T.}\right)}^{2}$ and I^2^ = $100\%*\frac{Q-df}{Q}$, where df is the degrees of freedom. A rough guide to interpretation of I^2^ is as follows: 0 to 40% might not be important, 30% to 60% may represent moderate heterogeneity, 50% to 90% may represent substantial heterogeneity, and 75% to 100% represents considerable heterogeneity [28]. We then performed a sensitivity analysis to decrease heterogeneity to less than 40%. During the sensitivity analysis, we excluded studies that caused the heterogeneity to be higher. Then we calculated the standard deviation (SD) for each weighted average as, $SD=\sqrt{N}*\frac{\left(upper\;limit-lower\;limit\right)}{3.92}$. Finally, we compared the weighted averages within and between the groups by assessing 95% CI of the differences of weighted averages calculated as ${\bar{x}}_{1}-{\bar{x}}_{2}\pm 1.96*\sqrt{\frac{\sigma_{1}^{2}}{n_{1}}+\frac{\sigma_{2}^{2}}{n_{2}}}$, where x_1_ and n_1_ are the mean and size of the first sample, and σ_1_ the first population’s SD, and x_2_ and n_2_ are the mean and size of the second sample, and σ_2_ the second population’s SD. In this case, if an interval does not contain zero, the corresponding means are significantly different. All the statistic and plots were carried out using Microsoft Excel 2016.

## 3. Results

### 3.1. Description of Studies and Study Population

Of the 1998 papers identified as potentially relevant during the database search, 19 were included. From the search of reference lists, we added 2 papers. Altogether 21 papers were finally analyzed (Figure 1). The quality of the included studies was sufficient according to the JBI-QARI score. Papers included data from 647 junior players (mean age 10.0–19.3) from 27 groups, and 277 senior players (mean age 20.7–27.1) from 8 groups. All the included groups came from teams that played at elite or sub-elite level. Studies published in the papers were conducted in the United Kingdom (4×) [29,30,31,32], Spain (4×) [33,34,35,36], Brazil (2×) [37,38], Italy (2×) [39,40], Netherlands (2×) [41,42], Australia (1×) [43], Greece (1×) [44], Mexico (1×) [45], Malta (1×) [46], Turkey (1×) [47], India (1×) [48], and France (1×) [49]. Several methods were used to record the dietary intake. Most of them were based on self-report or self-record approaches. Recording time varied from 24 h to seven consecutive days. A summary of studies describing the information is presented in Table 2.

### 3.2. Dietary Intake in Individual Groups

In junior players, the lowest energy intake per day was 1903 (432) kcal/day in 12.7 (0.6) year old players in the study by Naughton [31] and the highest energy intake was 3478 (223) kcal/day in 16.6 (0.2) year old players in the study by Ruiz [36]. In senior players, the lowest energy intake was 2164 (498) kcal/day in 27.1 (4.2) year old players in the study by Bonnicci [46] and the highest energy intake was 3442 (158) kcal/day in 24.8 (5.5) in the study by Hassapidou [44]. If divided according to the body mass, the lowest energy intake was 28.1 (6.8) kcal/kg/day in 16.4 (0.5) year old players in the Naughton study [31] and the highest energy intake was 77.8 (7.6) kcal/kg/day in 10.0 (0.8) year old players in the Cherian study [48] in juniors. The lowest energy intake per body mass per day was 29.7 (7.3) kcal/kg/day in senior players, aged 27 (5) years, in Devlin (2016) [43], and 46.0 (2.1) kcal/kg/day in Hassapidou [44] was the highest energy intake in seniors. The lowest protein intake per body mass per day was 1.3 (0.2) g/kg/day in 16–19-year-old players in the study by Murphy [30], while the highest protein intake was 2.2 (0.5) g/kg/day in 12.7 (0.6) year old players in the Naughton study [31]. There was also 2.2 g/kg/day found in three other studies amongst juniors. There was a very homogeneous result in protein intake per day in seniors. The lowest intake was 1.8 (0.1) g/kg/day in the Ruiz study [36], and 2.0 (1.3) g/kg/day was the highest intake in the Hassapidou study [44]. In contrast to protein, the CHO intake was heterogenic in both the groups. The lowest intake was 3.2 (1.3) g/kg/day in 16.4 (0.5) years old players in the Naughton study [31], and 12.9 (3.0) g/kg/day was the highest intake in 10-year-old players in the Cherian study [48]. A low intake of 2.9 (1.1) g/kg/day was found in senior players, aged 27 (5) years, in the Devlin study [43], which was the lowest intake in both groups. The highest intake in seniors was 5.9 (0.7) g/kg/day CHO in players aged 23 (7) years in the study by do Prado [37]. The lowest proportion of fat on TEI was 24(2)% in 10.0 (0.8) year old players in the study by Cherian [48]. The highest percentage was 39(1)% in 16.6 (0.2) year old players in the study by Ruiz [36]. Amongst seniors, there was 3 × 29% as the lowest percentage of fat on TEI a day and 41(6)% was the highest percentage in Hassapidou [44]. An overview of individual intake is given in Table 3.

### 3.3. Meta-Analysis

During sensitivity analysis, we excluded the groups Ruiz [36] (51.5 (2.5)) and Galanti [40] from the overall analysis, Murphy [30] from 2000–2009 analysis, and both the Cherian [48] groups from 2010–2019 analysis of energy intake. We excluded Murphy [30] from the overall as well as from 2000–2009 analyses and Hidalgo y Terán Elizondo [45] (2.2 (0.1)) from 2010–2019 analysis of proteins intake. We excluded Murphy [30] from the overall as well as from 2000–2009 analyses and both the Cherian [48] groups from 2010–2019 analysis of CHO intake. Finally, two of the Ruiz [36] groups (38 (2), 39 (1)) from the overall analysis and Ruiz [36] (39 (1)) from the 2000–2009 analysis and again both the Cherian [48] groups from 2010–2019 analysis of fat analysis were excluded. We did not exclude any study during the sensitivity analysis of senior players. Despite this, the I^2^ was 0% in almost all the senior players’ analyses except in the fat analysis where I^2^ was 20.5% in the overall analysis and 39.4% in the 2000–2009 analysis. Taking into account the reference value, protein intake was significantly higher than the upper level of the reference value (1.7 g/kg/day) in juniors, and CHO intake was significantly lower than the lower limit of the reference value (6 g/kg/day) in almost all the analyses except the juniors’ in 2000–2009. Percentage of fat on TEI showed normalized percentages in both groups. Forest plots with marked reference values showing each weighted average are presented for juniors in Figure 2 and for seniors in Figure 3.

Energy intake was very similar in both groups 44.1 (95% CI 42.5–45.7) in juniors and 42.9 (95% CI 40.1–45.6) kcal/kg/day in seniors. A decreasing tendency between 2000–2009 and 2010–2019 was also found in both groups; however, a statistically significant difference was only found in juniors. When the energy intake between juniors and seniors was compared over time, there was no statistically significant difference in energy. A very consistent result was found for protein intake in both groups. The protein intake oscillated between 1.8 to 2.0 g/kg/day in juniors and 1.8 to 1.9 g/kg/day in seniors, respectively. There was a statistically significant decreasing tendency between 2000–2009 and 2010–2019 in CHO intake in both categories. A significantly lower CHO intake was found in seniors compared to juniors at all times. Percentage of fat in TEI was very similar in both groups in overall analyses. This also decreased over time in both groups. The decrease between times was statistically significant in juniors. Table 4 shows the results of meta-analysis.

## 4. Discussion

The results of this meta-analysis show that there is a relatively stable development in protein intake in the junior as well in the senior category. Furthermore, there is a decreasing CHO intake trend amongst the junior as well as the senior players. In both categories, CHO intake is not adequate taking into account the recommended values. Despite the decreasing trend, the fat proportion in TEI is adequate in both categories.

Energy intake per body mass decreased in the last decade when compared with the first ten years of the 21st century. This estimate is interesting when considering the physiological demands of the soccer game, which have always been high and have increased substantially over the last decade [50,51,52]. In contrast, a relatively high protein intake in both categories is expected. A high protein intake has been recommended for athletes for many years [16,53,54,55,56]. Although evidence of a high protein diet causing harm in athletes has not been confirmed [57], there is compelling evidence that an intake in excess of approximately 1.7 g/kg/day will not help to build and repair muscles [58]. However, it should be noted that there is preliminary evidence that consuming much higher quantities of protein (>3 g/kg/day) may confer a benefit as it relates to body composition. Concerns that protein intake within this range is unhealthy are unfounded in healthy, exercising individuals [24].

While an increased protein intake is understandable, the lower CHO intake raises questions. It is well-known that an adequate CHO intake may play an important role in the recovery process [20,59,60]. CHO supplementation has shown to be a beneficial ergogenic aid in improving performance in soccer [61]. One of the important roles of CHO from diet in a human body is muscle glycogen resynthesizes before, during, and after sport activities. Muscle glycogen is the predominant energy source for soccer match play [61]. The game performance in soccer places great demands at muscle glycogen storage. At the end of the game, about half of the muscle fibers may be almost empty or empty of glycogen [62]. Development of fatigue during prolonged intermittent exercise has been associated with lack of muscle glycogen, and it has been demonstrated that elevating muscle glycogen prior to exercise through a CHO diet elevates performance during such a type of exercise [63]. CHO intake of less than 5 g/kg/day in the habitual diet of soccer players might be sufficient to replenish the muscle glycogen utilized during soccer specific performance. However, cumulative deficits of about 10% in glycogen replenishment might provoke decrements in performance [64]. Before an important soccer match, soccer players should have a high CHO intake that provides at least 8 g/kg/day CHO diet, and in every possible way should avoid eating a low 3 g/kg/day CHO diet before the match [60]. There is evidence that increasing the CHO intake from 5.4 to 8.5 g/kg/day allowed better maintenance of physical performance and mood state over the course of training in trained runners [65] and a range of evidence to improve the soccer performance after using different CHO supplementation before a game [61,66,67,68,69,70,71]. Moreover, a high CHO diet may be particularly important where there is limited recovery time between games [72]. It is not possible to conclude what causes players to cut CHO intake. It may have come about from as a consequence of warnings about CHO overuse in the general population. However, nutrition specialists who are responsible for players’ diets, particularly for seniors, should be aware of the facts and not discourage players from increasing CHO intake unnecessarily. The long-term low-carbohydrate regimens are potentially harmful to the performance of the athlete [73]. Among the problems mentioned, there are impaired cognitive performance, mood, perceptions of fatigue [65], and an inability to focus on the task or a greater susceptibility to skeletal muscle damage while training or competing [74]. Problems may also be in a low-carbohydrate with high-fat diet, which may result in a decrement in performance gains [75]. Hassapidou [44] and Ruiz [36] may probably be counted among the problematic ones in this case. Nevertheless, we did not find any trend that would be a concern for the other studies in this question.

Regarding the fat intake, in our meta-analysis this topic is relatively limited. Majority of studies included here have described just the fat intake in general as percentage of TEI. Differentiating between saturated and unsaturated fatty acids would be more appropriate. What is interesting is the decreasing trend in the proportion of fat intake between the first and second decades. Mean fat intakes noted by Iglesias–Gutiérrez [35], Ruiz [36], or Hassapidou [44] were above 35%; however, in later studies the proportion was about ≤30%. Although no mean proportion was below the recommended lower limit, which is 20% of TEI, it may be important to be aware of the trend in decreased fat consumption over recent decades. Considering the importance of fats in humans, we can conclude that the fat proportion in TEI is in concordance with the recommendations.

### Limitations

This meta-analysis has several limitations. Even though the quality of included studies was sufficient, the methods used to estimate players’ diet differed across studies. There were relatively high differences among the methods, recording season, or recording periods, which were used in the studies. Sometimes, the records were collected in consecutive days, whereas in other studies records were collected only on selected days, such as rest days, training days, pre-match days, and match days. Macronutrients were taken as general without regard of the quality in all the studies. Although all the groups came from clubs playing in elite or sub-elite leagues and nutrition specialists probably created diet recommendation for almost all the players who were included, national eating habits might have influenced results. For example, Malta, India, Turkey and Australia, which are not typically soccer-playing countries, were included only in the second decade.

## 5. Conclusions

In conclusion, taking into account the aforementioned limitations, this is the first meta-analysis that gives a comprehensive insight into the development of a macronutrient diet in soccer players during the first two decades of the 21st century. While protein intake is higher than the recommended in both seniors and juniors, it appears that despite recommendations, CHO intake is still below the recommended values in both groups. To avoid possible health problems the CHO intake should be more carefully monitored in soccer players. Responsible people (coaches, nutrition specialists, parents) and of course the players themselves should care to notice players’ normal daily intake and perhaps increase basal levels of CHO intake or daily intakes when replenishment of daily deficits are necessary. When possible, it may be beneficial to assess amounts of glycogen depletion to better understand the extent of deficits and how much replenishment may be necessary. The proportion of fat as TEI is in concordance with the recommendations. Future studies should consider using consistent methodology to improve the knowledge of nutritional trends in soccer players.

## Figures and Tables

**Figure 1 nutrients-11-01305-f001:**
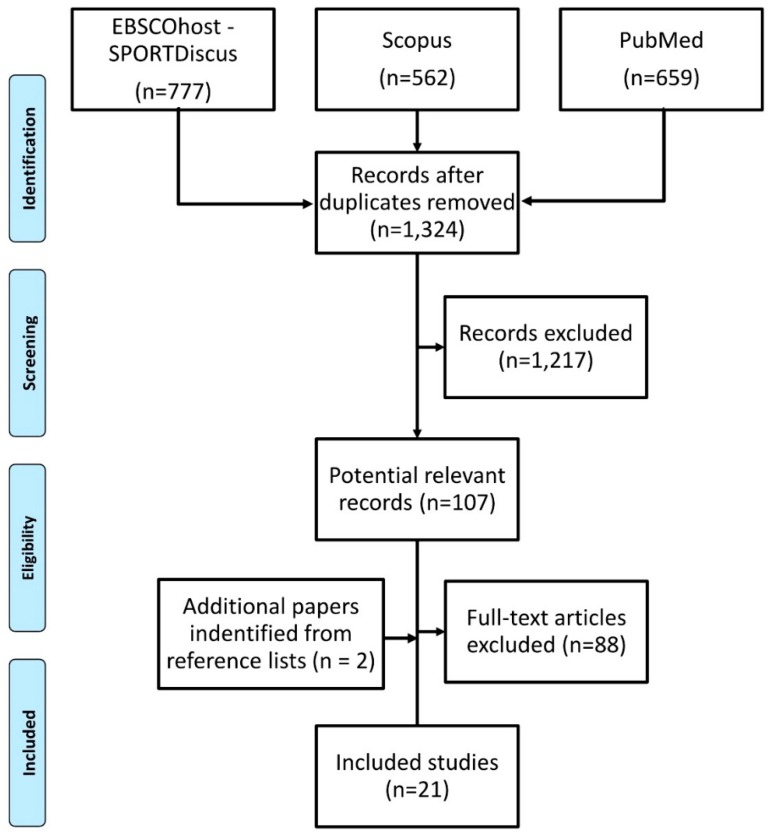
Flowchart illustrating the different phases of the search and study selection.

**Figure 2 nutrients-11-01305-f002:**
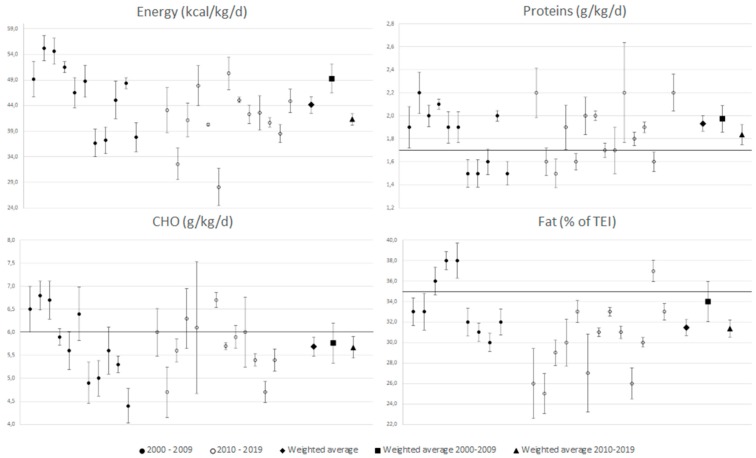
The forest plot for juniors. There are presented studies from 2000–2009 on the left side and from 2010–2019 on the middle and the weighted averages on the right side. Black line demonstrates higher level of recommended value in proteins and fat and lower level in carbohydrate (CHO). If 95% confidence interval (CI) does not cross the line, the result is statistically significant.

**Figure 3 nutrients-11-01305-f003:**
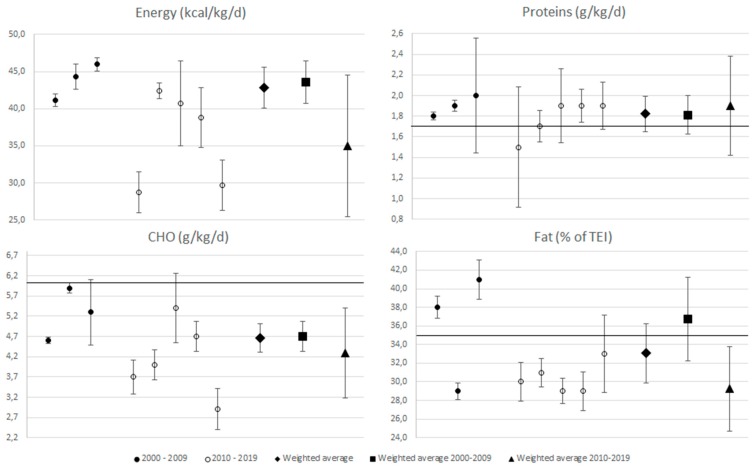
The forest plot for seniors. There are presented studies from 2000–2009 on the left side and from 2010–2019 on the middle and the weighted averages on the right side. Black line demonstrates higher level of recommended value in proteins and fat and lower level in CHO. If 95% CI does not cross the line, the result is statistically significant.

**Table 1 nutrients-11-01305-t001:** Search string and results from electronic databases.

DATABASE	KEY	NUMBER
EBSCOhost—SPORTDiscus	(soccer OR football) AND (nutrition OR diet OR macronutrient) Limiters: Published Date: 20000101-20191231; Source Types: Academic Journals	777
Scopus	((TITLE-ABS-KEY (soccer) OR TITLE-ABS-KEY (football))) AND ((TITLE-ABS-KEY (nutrition) OR TITLE-ABS-KEY (diet) OR TITLE-ABS-KEY (macronutrient))) AND (LIMIT-TO (PUBYEAR, 2019–2000) AND (LIMIT-TO (DOCTYPE, “ar”))	562
PubMed	Search (((soccer) OR football)) AND (((nutrition) OR diet) OR macronutrient) From 2000/01/01 to 2019/12/31	659

**Table 2 nutrients-11-01305-t002:** Descriptive information of the studies included in the analyses.

Study	Year	Country	Level	Recording Season	Recording Period (Days)	Note
Bettonviel [41]	2016	Netherlands	Dutch Premier League (Eredivisie)	The first half of the competitive season	4 nonconsecutive days (1 match day, 1 post match day, 1 rest day, 1 training day)	Self-reported, 24-h web-based recall method
Bonnicci [46]	2018	Malta	Malta BOV Premier League	The end of the competitive season	4 consecutive days (Monday to Thursday)	Self-reported, normal habitual diet
Briggs [29]	2015	UK	English Premier League	The second half of the competitive season	7 nonconsecutive days (4 training days, 2 rest days, 1 match day)	Self-reported, weighed food diary and 24-h recall
Brinkmans [42]	2019	Netherlands	Dutch Premier League (Eredivisie)	November until April	24-h × 3	Three unannounced face-to-face 24-h dietary recalls for a match-, training, and rest day
Caccialanza [39]	2007	Italy	Italian First Division	-	4 consecutive days (2 training days, 1 match day, 1 rest day)	Self-reported food records on a time zero (T0) and after 3 months (T1)
Devlin [43]	2016	Australia	Australian A-league	The end of the pre-competition period	24-h (training day)	Multiple-pass dietary recalls for a ‘training day’ during the end of preseason period
do Prado [37]	2006	Brazil	Brazilian Campeonato Paulista Série A1	The whole competitive season	-	Self-reported food consumed more than three times per week regardless whether the athlete was or not on a match day
Ersoy [47]	2019	Turkey	Pro-professional soccer	The pre-competition period	3 consecutive days	Three-day food consumption registration form
Galanti [40]	2015	Italy	Italian First Division	The competitive season	-	Questions about the type of food, the portions and frequency of consumption of various foods and beverages. The quantification of food intake was undertaken using pictures of food depicted in different portions with known weights from a dedicated photographic archive.
Garrido [33]	2007	Spain	Spain División de Honor Juvenil	The second half of the competitive season	5 consecutive days (4 weekdays and a weekend match day)	Self-reported; food intake was assessed on 2 occasions “buffet-style” diet and a fixed “menu-style” diet
Hassapidou [44]	2000	Greece	Elite Greek team	The competitive season	3 consecutive days (including the match day)	Self-recorded weighed dietary record
Hidalgo [45]	2015	Mexico	Mexican National Soccer League	The competitive season	4 consecutive days (excluding the match day)	Self-recorded weighed dietary record
Cherian [48]	2018	India	Junior national-level	-	3 consecutive days	Self-recorded weighed dietary record
Iglesias-Gutiérrez [34]	2012	Spain	Spanish First Division	The first half of the competitive season	6 consecutive days (a whole week, excluding the match-day)	Self-recorded weighed dietary record
Iglesias-Gutiérrez [35]	2005	Spain	Spanish First Division	The first half of the competitive season	6 consecutive days (a whole week, excluding the match-day)	Self-recorded weighed dietary record
Leblanc [49]	2002	French	French National Training Centre	The beginning of the year	5 consecutive days (including 3 weekdays and 2 weekend days)	Self-recorded using household measures such as cups, dishes, and spoons
Murphy [30]	2006	UK	English Premier League and a League One	The mid-season	7 consecutive days	Self-reported; to eliminate the possibility of under-reporting, the basal metabolic rate (BMR) of all participants was calculated. Any subjects thought to be underreporting were eliminated from the study.
Naughton [31]	2016	UK	English Premier League	-	7 consecutive days	Self-recorded
Raizel [38]	2017	Brazil	Brazilian professional soccer players	The pre-season	3 nonconsecutive days (2 nonconsecutive weekdays and a weekend day)	Semi-structured food record in pre-season
Ruiz [36]	2005	Spain	Spain Tercera División	-	3 consecutive days (Sunday, Monday, and Tuesday)	Self-recorded with a weighing scale and a questionnaire to record the type and quantity of food eaten.
Russell [32]	2011	UK	UK Championship soccer team	The first half of the competitive season	7 days (a match day, 4 training days, and 2 rest days in the first half of the competitive season)	Self-recorded using household measures such as cups, dishes, and spoons

**Table 3 nutrients-11-01305-t003:** Macronutrient intake in separate groups.

Study	Year	Age	*n*	Energy (kcal/day)	Energy (kcal/kg/day)	Proteins (g/kg/day)	CHO (g/kg/day)	Fat (% of TEI)
Bettonviel [41]	2016	17.3 (1.1)	15	2938 (465)	42.6 (6.7)	1.7 (0.4)	6.0 (1.5)	26 (3)
Briggs [29]	2015	15.4 (0.3)	10	2237 (320)	41.2 (5.3)	1.5 (0.2)	5.6 (0.4)	29 (2)
Caccialanza [39]	2007	16 (1)	43	2560 (636)	36.7 (9.1)	1.5 (0.4)	4.9 (1.5)	31 (3)
		16 (1)	43	2640 (614)	37.2 (8.7)	1.5 (0.4)	5.0 (1.3)	30 (3)
Ersoy [47]	2019	16 (1.2)	26	3225 (692)	47.9 (10.2)	1.9 (0.5)	6.3 (1.7)	30 (6)
Galanti [40]	2015	15–16	30	2844 (51)	40.3 (0.7)	1.6 (0.2)	6.1 (0.4)	33 (3)
Garrido [33]	2007	16.9 (1.5)	33	2740 (531)	37.8 (8.3)	1.5 (0.3)	4.4 (1.1)	-
		16.1 (1.4)	29	3148 (619)	45.1 (10.0)	1.6 (0.3)	5.6 (1.4)	-
Hidalgo [45]	2015	15.5 (0.0)	24	3067 (151)	50.3 (8.0)	2.2 (0.1)	6.7 (0.4)	31 (0)
		16.5 (0.0)	24	2930 (73)	45.1 (1.4)	2.0 (0.1)	5.7 (0.2)	33 (0)
		17.3 (0.0)	18	2715 (131)	40.7 (2.0)	1.9 (0.1)	5.4 (0.3)	30 (0)
		19.3 (0.2)	6	3042 (117)	44.9 (2.9)	2.2 (0.2)	5.4 (0.3)	33 (0)
Cherian [48]	2018	10.0 (0.8)	10	2871 (279)	77.8 (7.6)	2.2 (0.7)	12.9 (3.0)	24 (2)
		13.3 (1.5)	11	3237 (303)	59.6 (5.6)	1.8 (0.1)	9.3 (0.7)	26 (2)
Iglesias-Gutiérrez [34]	2012	18 (2)	87	2794 (526)	38.5 (8.5)	1.6 (0.4)	4.7 (1.1)	37 (5)
Iglesias-Gutiérrez [35]	2005	14–16	33	3003 (437)	46.5 (8.6)	1.9 (0.4)	5.6 (1.2)	38 (5)
Leblanc [49]	2002	13	19	2436 (374)	49.2 (7.6)	1.9 (0.4)	6.5 (1.1)	33 (3)
		14	19	2916 (286)	55.2 (5.4)	2.2 (0.4)	6.8 (0.7)	33 (4)
		15	19	3010 (427)	48.8 (6.9)	1.9 (0.3)	6.4 (1.3)	32 (3)
Murphy [30]	2006	16–19	22	2452 (430)	33.9 (5.9)	1.3 (0.2)	4.3 (0.3)	32 (3)
Naughton [31]	2016	12.7 (0.6)	21	1903 (432)	43.1 (10.3)	2.2 (0.5)	6.0 (1.2)	26 (8)
		14.4 (0.5)	25	1927 (317)	32.6 (7.9)	1.6 (0.3)	4.7 (1.4)	25 (5)
		16.4 (0.5)	13	1958 (390)	28.1 (6.8)	2.0 (0.3)	3.2 (1.3)	27 (7)
Ruiz [36]	2005	14.0 (0.1)	18	3456 (309)	54.6 (5.5)	2.0 (0.2)	6.7 (0.9)	36 (3)
		14.9 (0.5)	20	3418 (182)	51.5 (2.5)	2.1 (0.1)	5.9 (0.4)	38 (2)
		16.6 (0.2)	19	3478 (223)	48.4 (2.4)	2.0 (0.1)	5.3 (0.4)	39 (1)
Russell [32]	2011	17 (1)	10	2831 (164)	42.3 (2.9)	1.7 (0.1)	5.9 (0.4)	31 (1)
**Seniors**								
Bettonviel [41]	2016	22.8 (3.7)	14	2988 (583)	38.8 (7.6)	1.9 (0.3)	4.7 (0.7)	29 (4)
Bonnicci [46]	2018	27.1 (4.2)	22	2164 (498)	28.7 (6.6)	1.5 (0.4)	3.7 (1.0)	30 (5)
Brinkmans [42]	2019	23 (4)	41	3285 (354)	42.4 (3.5)	1.7 (0.5)	4.0 (1.2)	31 (5)
Devlin [43]	2016	27 (5)	18	2247 (550)	29.7 (7.3)	1.9 (0.5)	2.9 (1.1)	33 (9)
do Prado [37]	2006	23 (7)	118	3371 (721)	44.3 (9.5)	1.9 (0.3)	5.9 (0.7)	29 (5)
Hassapidou [44]	2000	24.8 (5.5)	21	3442 (158)	46.0 (2.1)	2.0 (1.3)	5.3 (1.9)	41 (5)
Raizel [38]	2017	20.7 (2.0)	19	2924 (460)	40.7 (12.8)	1.9 (0.8)	5.4 (1.9)	29 (3)
Ruiz [36]	2005	20.9 (0.5)	24	3030 (141)	41.1 (2.1)	1.8 (0.1)	4.6 (0.2)	38 (3)

Note: Data is presented as mean and standard deviation (SD).

**Table 4 nutrients-11-01305-t004:** Weighted averages and differences between juniors and seniors.

	Junior Players	Heterogeneity	Senior Players	Heterogeneity
	Weighted Average (95% CI)	I^2^	Weighted Average (95% CI)	I^2^
Energy (kcal/kg/day)	44.1 (42.5–45.7)	3.3	42.9 (40.1–45.6)	0
2000–2010	49.3 (46.5–52.1)	0	43.6 (40.7–46.4) ^†^	0
2010–2019	41.3 (40.2–42.4) *	22.2	35.0 (25.5–44.6)	0
Proteins (g/kg/day)	1.9 (1.8–2.0)	24.7	1.8 (1.6–2.0)	0
2000–2010	2.0 (1.9–2.1)	5.6	1.8 (1.6–2.0)	0
2010–2019	1.8 (1.7–1.9)	23.4	1.9 (1.4–2.4)	0
CHO (g/kg/day)	5.7 (5.5–5.9)	0	4.7 (4.3–5.0) ^†^	0
2000–2010	5.8 (5.3–6.2)	0	4.7 (4.3–5.1) ^†^	0
2010–2019	5.7 (5.4–5.9)	13.3	4.3 (3.2–5.4) ^†^	0
Fat (% of TEI)	31.5 (32.0–35.9)	4.4	33.1 (29.9–36.2)	20.5
2000–2010	34.0 (32.0–35.9)	0	36.7 (32.3–41.2)	39.4
2010–2019	31.4 (30.5–32.2) *	25.7	29.3 (24.7–33.8) *	0

Note: * Statistically significant difference from 2000–2010; ^†^ statistically significant difference between juniors and seniors; CHO—carbohydrates; TEI—total energy intake.

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
