# Peer review of "Macronutrient Intake in Soccer Players—A Meta-Analysis"

_nutrients, 2019, doi:10.3390/nu11061305_

Reviewer 1 Report

Very interesting manuscript. Please consider the following:

Line 9-18: Abstract should mention the important results of the meta-analysis regarding number of articles, statistics, groups, etc.

Line 18: Remove acronyms "CHO" and "TEI" since they are not used anymore in the abstract.

Line 19: "European football" is not a valid MeSH term. 

Line 28: Introduce the acronym CHO where corresponds.

Line 36: Repair and remodeling are part of protein turnover, try to use a term that encompass the context or use parenthesis.

Line 37: Introduce the acronym TEI where corresponds.

Line 27-37: Consider also including recommendations reported in the ISSN position stands:

Protein by Jäger et al. 2017

Nutrient timing by Kerksick et al. 2017

Research & recommendations by Kerksick et al. 2018

Line 48: Did you register the meta-analysis in PROSPERO?

Line 55: Include "search string" in the name of the table.

Line 86: Why t-test? There are more powerful and insightful statistical approaches 

Line 103: Preserve right units for hours (h) not hr. Please make necessary changes.

Line 112: Avoid stating sentences with a number.

Line 116, 117, 119: Idem 112

Line 119-123: Please use units (g/kg/day)

Line 121-122: It's important to remark the population for this value (2.9 [1.1] g/kg/day for seniors), even though Devlin et al. refers to this population.

Line 148: Both figures should have a better explanation of the represented results. 

Line 161: In discussion authors should also report the potential decremental effect of carbohydrates on performance. Also analyze data by countries (even though is mentioned in limitations) or first vs second division or professionals vs U20/U17/U12, etc.

Line 182 & 191: Limitations and conclusiones should be in a different section. 

Author Response

Dear reviewer,

We appreciate all your comments. We considered all the comments during revision of the manuscript and we believe that they contributed to improvement of its quality. We hope that the revised manuscript will be considered adequate for publishing in the Nutrients.

Our specific responses to your comments follow:

 Line 9-18: Abstract should mention the important results of the meta-analysis regarding number of articles, statistics, groups, etc.

Abstract has been updated to include the reviewer’s suggestions.

Line 18: Remove acronyms "CHO" and "TEI" since they are not used anymore in the abstract.

We have done it.

Line 19: "European football" is not a valid MeSH term.

Now we used the official name "Association football".

Line 28: Introduce the acronym CHO where corresponds.

We have done it.

Line 36: Repair and remodeling are part of protein turnover, try to use a term that encompass the context or use parenthesis.

We have used parentheses.

Line 37: Introduce the acronym TEI where corresponds.

We have done it.

Line 27-37: Consider also including recommendations reported in the ISSN position stands:

Protein by Jäger et al. 2017

Nutrient timing by Kerksick et al. 2017

Research & recommendations by Kerksick et al. 2018

We are grateful for these suggestions and thank the reviewer for the recommendation. We have added all of them and feel they strengthened the paper.

Line 48: Did you register the meta-analysis in PROSPERO?

No, we did not register our meta-analysis in PROSPERO. We thought that the main implication of this work would be on the sport performance. The PROSPERO inclusion criteria cover mainly healthy connected reviews. Nevertheless, we now know that such implications our analysis has. Unfortunately, ongoing work is not allowed to be registered.

Line 55: Include "search string" in the name of the table.

We have added "search string".

Line 86: Why t-test? There are more powerful and insightful statistical approaches

We used different approach to test hypothesis.

Line 103: Preserve right units for hours (h) not hr. Please make necessary changes.

We have done it.

Line 112: Avoid stating sentences with a number.

Line 116, 117, 119: Idem 112

We have avoided it.

Line 119-123: Please use units (g/kg/day)

We have used it.

Line 121-122: It's important to remark the population for this value (2.9 [1.1] g/kg/day for seniors), even though Devlin et al. refers to this population.

We have done it.

Line 148: Both figures should have a better explanation of the represented results.

We have added such information.

Line 161: In discussion authors should also report the potential decremental effect of carbohydrates on performance. Also analyze data by countries (even though is mentioned in limitations) or first vs second division or professionals vs U20/U17/U12, etc.

We have added information about the potential detrimental effect of carbohydrates. Nevertheless, despite wanting the analyses across countries or first vs second division or professionals vs U20/U17/U12 we cannot do that because of the a small number of studies and relatively high differences among the methods, recording season or recording periods which were used in the studies. We have added a few sentences about these problems in the Limitations section.  

Line 182 & 191: Limitations and conclusiones should be in a different section.

We have separated the Limitations section and Conclusions.

Reviewer 2 Report

Dear authors,

When I have read the paper I haven´t understood the objective of this study: "The aim of this meta-analysis was to uncover current dietary trends regarding three basic macronutrients in junior as well as in senior soccer players from elite and sub-elite leagues"

Why is interesting to realyze a meta-analysis for know the macronutrient intake? What is it the practical applications of the results? What is the relation cause effect in your scientific problem?

In the introduction authors don´t include the importance of their objective. Why is necessary to include a higher o lower intake of a macronutrient in soccer? All the intervention in sports nutrition can be realyzed attending to the limiting factor of performance in an athlete modality. So, in introduction authors must be explain it.

I think it was more interesting to analyze the effect of different studies realyzed in soccer manipulating the glycogen bioavailability attending to the intake of carbohydrates, for example.

I think that authors must change the objective of a future work in this temathic. Also, I recommend to describe the process of article selection and inscribe the review or meta-analysis in PROSPERO. Inscribing the meta-analysis in PROSPERO the process guarantee realyze an effective review and/or meta-analysis.

I want to aime authours to follor work and improving in the future.

Author Response

Dear reviewer,

We appreciate your comments.

 Dear authors,

When I have read the paper I haven´t understood the objective of this study: "The aim of this meta-analysis was to uncover current dietary trends regarding three basic macronutrients in junior as well as in senior soccer players from elite and sub-elite leagues"

The authors thank the reviewer for their comment and have made changes throughout the paper to better reflect the objective.

Why is interesting to realyze a meta-analysis for know the macronutrient intake? What is it the practical applications of the results? What is the relation cause effect in your scientific problem?

We think that nutrition plays an important role in normal growth in juniors and in reducing the risk of illness and injuries in older athletes. Our results may serve as an example for coaches to make diet recommendation for players. We have mentioned this in manuscript.

In the introduction authors don´t include the importance of their objective. Why is necessary to include a higher o lower intake of a macronutrient in soccer? All the intervention in sports nutrition can be realyzed attending to the limiting factor of performance in an athlete modality. So, in introduction authors must be explain it.

The introduction was updated to include such information.

I think it was more interesting to analyze the effect of different studies realyzed in soccer manipulating the glycogen bioavailability attending to the intake of carbohydrates, for example.

Good point we will do meta-analysis in this problem in the future. Thank you for a good type.

I think that authors must change the objective of a future work in this temathic. Also, I recommend to describe the process of article selection and inscribe the review or meta-analysis in PROSPERO. Inscribing the meta-analysis in PROSPERO the process guarantee realyze an effective review and/or meta-analysis.

No, we did not register our meta-analysis in PROSPERO. We thought that the main implication of this work would be on sport performance. The PROSPERO inclusion criteria cover mainly healthy connected reviews. Nevertheless, we now know that such implications our analysis has. Unfortunately, ongoing work is not allowed to be registered.

I want to aime authours to follor work and improving in the future.

We considered all the comments during revision of the manuscript and we believe that they contributed to improvement of its quality. 

Thank you very much for your comments.

Round  2

Reviewer 2 Report

Dear authors,

 I want to put in value the effort realyzed for response all my comments, but I lament that you only have responded to me and don´t realyzed the changes proposed. Nevertheless, I want to say you that in the future when you realyze changes in a manuscript, answering to reviewer, you must include the change across track changes. The revision of the change has been so difficult for it.

 I have the next comment:

 Line 27-28: macronutrients could have ergogenic effect, but they aren´t ergogenic aids.

Line 29: you have to define correctly the term “ergogenic aids” because reading the paragraph this concept is confused.

I think is necessary to include information about possible depletion of glycogen store in a soccer competition.

There aren´t reflected the changes that I requested about the improvement in the conceptualization of the objective. I´m continuous questioning that “Why is interesting to realyze a meta-analysis for know the macronutrient intake? What is it the practical applications of the results? What is the relation cause effect in your scientific problem?” This questions aren´t still responded.

I countinuos thinking that analyse the effect of different studies realyzed in soccer manipulating the glycogen bioavailability attending to intake of carbohydrates is necessary in this study.

The discussion is so poor and it don´t include the most important focus about this thematic.

Author Response

Dear reviewer,

We apologize that in the first round we did not satisfy your requirements. We may not have completely understood the comments, which led to confusion on our end. However, we now believe that the new version with changes will be better suited.

 You must include the change across track changes. The revision of the change has been so difficult for it.

We included the track changes version in separate file.

 I have the next comment:

Line 27-28: macronutrients could have ergogenic effect, but they aren´t ergogenic aids.

We have changed these sentences.

Line 29: you have to define correctly the term “ergogenic aids” because reading the paragraph this concept is confused.

We have better described this.

I think is necessary to include information about possible depletion of glycogen store in a soccer competition.

We have included it.

There aren´t reflected the changes that I requested about the improvement in the conceptualization of the objective. I´m continuous questioning that “Why is interesting to realyze a meta-analysis for know the macronutrient intake? What is it the practical applications of the results? What is the relation cause effect in your scientific problem?” This questions aren´t still responded.

We hope that it is now better described.

 I countinuos thinking that analyse the effect of different studies realyzed in soccer manipulating the glycogen bioavailability attending to intake of carbohydrates is necessary in this study.

We know that this topic is very important in soccer; unfortunately, we really cannot include intervention trials into meta-analysis in observation studies. However, we discussed this topic deeply in Discussion.

 The discussion is so poor and it don´t include the most important focus about this thematic.

We hope that the discussion is now much better.

 We thank you for all your comments.

Sincerely,

Authors

Round  3

Reviewer 2 Report

Dear authors,

 I appreciate that at least you have considered some of my comments and recommendation. I believe that this meta-analysis isn´t novelty and it has a very low interest for the scientific community. Actually, there is an important interest in the periodization of carbohydrates intake in athlethes and it cannot be accepted a study that not considered it. My opinion is that the only study of this type interesting actually is a study that analyse the comparative effect of different CHO intake.

 Despite the low interest of the study, the quality of it is too lower. So it´s necessary to include, at least, all the next comments:

-          Line 35-38: in this section you cannot include a justification of your study. The justification must be located at the final of introduction.

-          Line 44-45: it´s necessary to include a reference specific of soccer.

-          It´s necessary to include more information about the possible depletion of glycogen store in soccer. Also, it could be interesting to include more information about protein intake and their possible beneficial effects (actually is very limited the information about it).

-          In the final of the introduction sections is necessary to include better the conceptualization of the objective. It´s necessary to continue improving introduction section.

-          Line 232-240: it must be included only references of studies realyzed in soccer players.

-          Line 237-238: sufficient or not sufficient?

-          Reference 65 cannot be used because it isn´t specific of soccer.

-          The previous comment must be widespread all discussion section. The discussion must be realyzed only with specific literature.

-          It´s necessary to make reference to different diets that actually are realyzed for manipulating glycogen bioavailability.